# Soluble and Exosome-Bound α-Galactosylceramide Mediate Preferential Proliferation of Educated NK Cells with Increased Anti-Tumor Capacity

**DOI:** 10.3390/cancers13020298

**Published:** 2021-01-15

**Authors:** Arnika K. Wagner, Ulf Gehrmann, Stefanie Hiltbrunner, Valentina Carannante, Thuy T. Luu, Tanja I. Näslund, Hanna Brauner, Nadir Kadri, Klas Kärre, Susanne Gabrielsson

**Affiliations:** 1Department of Microbiology, Tumor and Cell Biology (MTC), Karolinska Institutet, SE-17165 Stockholm, Sweden; Valentina.carannante@ki.se (V.C.); Hanna.Brauner@ki.se (H.B.); Klas.Karre@ki.se (K.K.); 2Center for Regenerative Medicine (HERM), Department of Medicine Huddinge, Karolinska Institutet, SE-14186 Stockholm, Sweden; Thuy.luu.thanh@ki.se (T.T.L.); Nadir.Kadri@ki.se (N.K.); 3Division of Immunology and Allergy, Department of Medicine, Solna, Karolinska Institutet, SE-17176 Stockholm, Sweden; ulf.gehrmann@googlemail.com (U.G.); Stefanie.Hiltbrunner@usz.ch (S.H.); tanja_naslund@hotmail.com (T.I.N.); 4Division of Rheumatology, Department of Medicine, Solna, Karolinska Institutet, SE-17176 Stockholm, Sweden; 5Dermato-Venerology Clinic, Karolinska University Hospital, SE-17164 Stockholm, Sweden; 6Science for Life Laboratory, Division of Infectious Diseases, Department of Medicine Solna, Karolinska Institute, Karolinska University Hospital, Solna, SE-17176 Stockholm, Sweden

**Keywords:** extracellular vesicles, exosomes, natural killer (NK) cells, innate anti-tumor response, missing-self response, antitumor immunity, NK cell education, MHC class I

## Abstract

**Simple Summary:**

Many cancers down-regulate the expression of major histocompatibility complex class I (MHC-I) to avoid recognition and elimination by the adaptive immune system. Natural Killer (NK) cells, as part of innate immunity, complement tumor recognition by their ability to sense foreign and malignant cells by the lack of major MHC-I expression. During development, NK cells learn to perceive physiological levels of self MHC-I as healthy in a process termed NK cell education. In the current study, we assessed whether the stimulation of natural killer T (NKT) cells could impact the tumor-killing ability of NK cells. We can show that stimulation with a well-established activator of NKT cells induces the preferential expansion of educated NK cells, and improves NK-cell-mediated tumor-killing of both MHC-I^+^ and MHC-I^−^ tumor cells. Furthermore, this increased tumor-killing capacity of NK cells can be achieved by nanovesicle-mediated delivery, which is known to improve adaptive immunity as well. Our approach thus holds great potential for future cancer immunotherapy.

**Abstract:**

Natural killer (NK) cells can kill target cells via the recognition of stress molecules and down-regulation of major histocompatibility complex class I (MHC-I). Some NK cells are educated to recognize and kill cells that have lost their MHC-I expression, e.g., tumor or virus-infected cells. A desired property of cancer immunotherapy is, therefore, to activate educated NK cells during anti-tumor responses in vivo. We here analyze NK cell responses to α-galactosylceramide (αGC), a potent activator of invariant NKT (iNKT) cells, or to exosomes loaded with αGC. In mouse strains which express different MHC-I alleles using an extended NK cell flow cytometry panel, we show that αGC induces a biased NK cell proliferation of educated NK cells. Importantly, iNKT cell-induced activation of NK cells selectively increased in vivo missing self-responses, leading to more effective rejection of tumor cells. Exosomes from antigen-presenting cells are attractive anti-cancer therapy tools as they may induce both innate and adaptive immune responses, thereby addressing the hurdle of tumor heterogeneity. Adding αGC to antigen-loaded dendritic-cell-derived exosomes also led to an increase in missing self-responses in addition to boosted T and B cell responses. This study manifests αGC as an attractive adjuvant in cancer immunotherapy, as it increases the functional capacity of educated NK cells and enhances the innate, missing self-based antitumor response.

## 1. Introduction

Natural killer (NK) cells are cytotoxic effector cells with important functions in antitumor immunity. These cells were initially characterized for their natural ability to lyse tumor cells lacking major histocompatibility complex class I (MHC-I) [1,2]. During NK cell development in the bone marrow (BM), most NK cells start to express activating and inhibitory receptors, which allow them to recognize and eliminate cancers, virus-infected and stressed cells. The activating receptors bind to pathogen-encoded ligands or endogenous stress molecules, while most inhibitory receptors (Ly49 receptors in mice, Killer Ig-like receptors (KIR) in humans, CD94/NKG2A in both species) recognize MHC-I molecules. Upon down-regulation of MHC-I, e.g., in a tumor setting, educated NK cells lose their inhibitory input. Simultaneous activation by stress-induced molecules on target cells will ultimately lead to killing of the tumor cell [3]. NK cells that express inhibitory receptors and are able to bind self-MHC-I alleles acquire the ability to recognize target cells that have lost the particular MHC-I allele. This gain in functional capacity, e.g., killing of target cells or IFNγ secretion, is termed NK cell education [4,5,6,7,8], while the process of killing MHC-I^low/neg^ cells is termed missing self-response [1,9]. While NK cells can become educated through interactions of other inhibitory receptors with their respective ligands, in this study we focus on NK cells educated by MHC-I. We will use the term “uneducated NK cells” for cells that have not experienced an education process due to the binding of self-MHC-I. Other inhibitory receptors of NK cells include NKG2A and LIR-1 which bind to HLA-E or HLA-G, respectively, and the adhesion molecule KLRG1, which recognizes members of the cadherin family. In addition to mediating inhibition, KLRG1 is associated with NK cell education [9,10,11] and marks mature NK cells that have been activated by virus or cytokines [12]. During maturation, NK cells develop from CD27^−^CD11b^−^ immature NK cells, then first acquire CD27 and then CD11b, to then lose expression of CD27 while up-regulating KLRG1 [13,14].

Activating receptors recognize a variety of ligands, some that are constitutively expressed and some that are up-regulated during cellular stress [3,15]. The latter is exemplified by NKG2D, which binds to stress-induced ligands such as Rae-1γ and H60 in mice [16,17]. However, NK cells might also be activated as a bystander effect of an ongoing immune response. For instance, the glycolipid α-galactosylceramide (αGC) is commonly used to stimulate type I, invariant NKT (iNKT) cells, but has been shown to induce strong NK cell activation as well [18,19,20,21,22]. NKT cells show features of both innate and adaptive lymphocytes, as they express NK cell markers such as NK1.1 or NKp46 as well as T cell receptor (TCR). iNKT cells can be divided in two subsets based on the class of ligands they recognize via their TCR. Class I iNKT cells express a TCR that combines the Vα14 with selected Vβ chains (e.g., Vβ2, 7, 8) and typically recognize glycolipids such as αGC [23]. The TCRs of type II NKT cells are more diverse as they are not restricted to the Vα14 chain and can recognize sphingo-, glycerol- or phospholipids rather than glycolipids [24]. Both subsets are activated following presentation of their respective lipid antigens on the MHC-Ib molecule CD1d. CD1d is mostly expressed by antigen-presenting cells (APC), such as conventional dendritic cells (cDC) and B cells in secondary lymphoid organs where antigen presentation takes place [24,25]. Thus, iNKT-cell-mediated NK cell activation seems to be dependent on a crosstalk between iNKT cells and CD1d-expressing APCs [19,26,27,28,29]. Here, direct cell-to-cell interactions via CD28-CD80/CD86 and CD40/CD40L and soluble factors such as interleukins (IL)-12, -15 and -18 as well as the Th1 cytokines IFNγ and tumor necrosis factor (TNF) play important roles in the activation of DCs, as well as iNKT cells and NK cells [18,19,20,26,27,28,29,30,31]. While the activation of iNKT cells can enhance both innate and adaptive immune responses [22,32], previous studies report adverse effects such as liver injury with transiently elevated levels of liver enzymes in αGC-treated mice [20,28,33], possibly as result of an excessive immune activation. While it is known that iNKT-cell-dependent NK cell activation increases IFNγ production and the degranulation of CD27^high^ NKs [22,31,34], little is known about the differential impact on educated vs. uneducated NK cells and the ensuing antitumor response. Whether iNKT-cell-activated NK cells can become auto-reactive and attack MHC-I^low/neg^ cells has not been analyzed to date.

Exosomes are nano-sized extracellular vesicles (EVs) with functions in cellular communication, as they can transfer lipids, proteins and nucleic acids. Exosomes are generated in multivesicular bodies (MVB) and are released into the extracellular space after the fusion of MVBs with the plasma membrane [35]. Due to their immunogenic properties, including the ability to carry tumor antigens, exosomes from DCs were investigated as cancer immunotherapeutic agents, first in mice [36] and later in clinical trials [37,38,39]. However, results in humans only showed moderate immune stimulatory effects, notably on NK cells. Further understanding of their immunological effects in vivo could result in ways to increase their potency. DC-derived exosomes have been shown to activate NK cells [40,41,42], B cells, as well as CD4^+^ and CD8^+^ T cells, in an antigen-specific manner in vitro and in vivo [36,42,43,44,45,46]. Interestingly, DC-derived exosomes express CD1d, and can be loaded with αGC, which leads to a significant boost in the exosome-induced adaptive immune response [47]. Exosomal αGC has an advantage over soluble αGC, as repeated injections, often needed in cancer therapy, of exosome-bound αGC did not induce the NKT cell anergy seen with soluble αGC. Strikingly, exosomal and soluble αGC induced a potent bystander NK cell activation [47], which led us to investigate the nature of this response in the current study. Given the importance of NK cells educated against self-MHC-I in antitumor immunity [48,49,50], we asked whether we could tilt this response in favor of antitumor NK responses.

Here, we report that soluble and exosomal αGC induce a preferential proliferation of educated NK cells in mice, and that this bias is strictly dependent on the MHC-I background of the mouse strain. Importantly, immunization with (exosomal) αGC significantly boosted the missing self-response. The activation of both educated and uneducated NK cells leads to an increased rejection of cancer cells without a breaking of tolerance towards healthy autologous cells. This finding may be important for the design of future cancer immunotherapy, where the down-regulation of self-MHC-I has been identified as a tumor escape mechanism [51,52]. In addition, it argues for the possibility of using αGC and other antigens loaded on exosomes to enhance both innate and adaptive antitumor immunity.

## 2. Results

### 2.1. αGC induces Strong NK Cell Activation and Proliferation

In recent years, it has become evident that NK cells are heterogeneous cells with a number of functional subsets that respond differently in different contexts. However, it remains unclear how different, unspecific modes of activation affect the balance between educated versus uneducated NK cells. To address this, we assessed the NK cell response to a strong immunogenic agent, αGC, known to induce iNKT cell activation and a marked bystander NK cell activation [18,19,20,22,31,34,47]. As reported previously, αGC induced NK cell activation and proliferation in the first 7 days post treatment (Figure 1A as evidenced by increased Bromodeoxyuridine (BrdU) incorporation and the expression of activation markers CD69, chemokine receptors CXCR4, CX3CR1 and CCR5, and decreased CD62L expression (Appendix A). While the frequency of splenic NK cells decreased, absolute NK cell numbers increased due to an increase in total lymphocytes in the spleen of treated mice (Appendix A). Interestingly, at day 3 post-treatment, αGC treatment led to a strong increase in expression of the maturation marker KLRG1. The increase in KLRG1 in mice stimulated with αGC was considerably higher than in mice treated with polyinosinic:polycytidylic acid (polyI:C) (Figure 1C), a dsRNA-analogue commonly used to activate NK cells via bystander activation [53]. The αGC-induced increase in KLRG1 intensity was MHC-I-independent but CD1d-dependent (Figure 1C). Up-regulation of KLRG1 on NK cells was most notable on maturing, CD27^hi^/CD11b^−^ and CD27^hi^/CD11b^+^ NK cells (Figure 1D), which were also increased in number as previously reported [31]. This indicates that αGC-induced NK cell activation is qualitatively different than polyI:C-induced activation and leads to functionally more mature NK cells [12,14,54].

### 2.2. Treatment with αGC Increases KLRG1^+^ NK Cell Function

To assess whether αGC also enhanced functional maturation, WT or CD1d^−/−^ mice were pre-treated with PBS or αGC in vivo before NK cell degranulation and IFNγ production was measured following ex vivo re-stimulation. As expected, we observed an increase in both CD107a expression (degranulation) and IFNγ expression following αGC-pre-treatment (Figure 2A–E). This effect was further enhanced by stimulating NK cells in vitro with αNK1.1 antibody. All αGC-induced effects were strictly dependent on the presence of iNKT cells during αGC treatment, as the effects of αNK1.1-treatment on IFNγ and CD107a expression were not seen in NK cells from CD1d^−/−^ mice (Figure 2B,C). Since KLRG1^+^ NK cells are important for tumor cell killing in vivo [54], we asked whether αGC treatment also enhanced the functional capacity of this subset. Indeed, KLRG1^+^ NK-cells from αGC-treated WT mice were significantly more potent in their degranulation and cytokine response following αNK1.1 stimulation than either their KLRG1^−^ counterparts or KLRG1^+^ NK cells from PBS-treated mice (Figure 2D,E). In order to visualize high-dimensional single-cell data, we used the *t*-distributed stochastic neighbor embedding (*t*-SNE) algorithm for non-linear dimensionality reduction [55], where the positions of cells reflect their proximity in high-dimensional space (Figure 2F–H). Combining flow cytometry data of mice from different groups (B6, B6 + αGC) we were able to identify NK cell populations that respond to the treatment (Figure 2F). Areas which showed 10% enrichment of cells in αGC-treated mice (as compared to PBS-treated mice) were defined manually (Figure 2F) and, thereafter, applied to mice from different treatment groups (Figure 2G). In accordance, t-SNE analysis of αNK1.1-restimulated NK cells illustrates that αGC-treatment had a larger effect in B6 as compared to CD1d^−/−^ mice (Figure 2G). Deeper analysis of individual markers revealed that αGC-induced changes correlated with KLRG1 expression (Figure 2H), while CD107 and IFNγ were increased even in cells that did not show the strongest response to treatment (Appendix A). Expression patterns of inhibitory receptors NKG2A and Ly49C correlated with αGC treatment (Appendix A). The expression of Ly49I, Ly49G2 and Ly49A only partially overlapped with cells that were most affected by αGC treatment (Appendix A). We conclude that stimulation with αGC not only increases the pool of KLRG1^+^ NK cells but also enhances their function with respect to degranulation and cytokine production.

### 2.3. αGC-Induced NK Cell Activation Is Partially DC-Dependent

Given the importance of conventional DCs (cDC), in particular CD8α^+^ cross-presenting cDCs1, as APCs for iNKT cell activation [19,26,27,28,29,32], and for maintaining NK cell functions [56,57,58], we asked whether DCs were important for αGC-induced NK cell activation. Here, in vitro activation of splenocytes with increasing doses of αGC led to a dose-dependent increase in CD69 on the surface of NK cells 24 and 48 h after activation (Figure 3A and Appendix A). Using co-cultures of ex vivo isolated WT NK cells and iNKT cells with DCs from WT B6 or CD1d^−/−^ mice, we could establish that the expression of CD1d on DCs was necessary to induce NK cell activation following αGC stimulation (Figure 3B). To confirm the role of DCs in αGC-mediated NK cell activation in vivo, we first measured the expression of DC activation markers following αGC injection. Treatment with αGC led to a significant increase in CD80 and CD40 expression in CD11c^high^CD11b^low^CD8α^+^ cDCs (gating strategy in Appendix A) from B6 and MHC-I^−/−^ and a trend towards an increase in CD86, but not from CD1d^−/−^ mice, 48 h after injection (Figure 3C), similar to a previous report [28]. Interestingly, the MHC-I molecule H2-Kb was slightly, yet insignificantly increased on cDC1s following αGC treatment in WT, but not in CD1d^−/−^ mice (Figure 3D). Thus, αGC-immunization leads to iNKT-cell dependent cDC1 activation in vivo. To test whether DCs were necessary for NK cell activation in vivo, we depleted DCs 24 h prior to treatment with αGC using the CD11c.DOG mice, genetically engineered to express the Diphteria toxin receptor under CD11c-promotor control. Following injection of Diphtheria, toxin DCs are depleted from CD11c.DOG, but not WT, mice [57]. We found that αGC-injection in DC-depleted animals induced reduced NK cell proliferation, as measured by BrdU incorporation and activation, measured as the percentage of KLRG1^+^ NK cells, as compared to WT controls (Figure 3E,F). DC depletion also affected αGC-induced changes in proportions of (im)mature NK cells, as measured by expression of CD11b and CD27 (Appendix A). We concluded that DCs are necessary for αGC-induced NK cell activation in vitro and for full NK cell activation in vivo.

### 2.4. αGC Induces Preferential Proliferation and Activation of Educated NK Cells

Because of the known association between KLRG1 and NK cell education [9,10,11], we examined whether the presence or absence of MHC-I had an impact on αGC-induced NK cell activation. The injection of αGC led to an increase in KLRG1^+^ NK cells in both B6 WT and MHC-I^−/−^ mice but not in CD1d^−/−^ mice (Figure 1C), suggesting that αGC can upregulate KLRG1 in the absence of MHC-I. However, it remained unclear whether αGC has a differential effect on educated NK cells and the missing-self response [9]. NK cells express a combination of different Ly49 receptors, each of which can bind to one or several MHC-I alleles. The inhibitory receptors, Ly49C and Ly49I receptors, bind to the MHC-I allele K^b^ in mice with H-2^b^ haplotype (K^b^D^b^), while Ly49A and Ly49G2 bind to D^d^ of H-2^d^ mice. NKG2A binds to HLA-E in most mouse strains. As a consequence, NK cells expressing Ly49C, Ly49I or NKG2A are educated in K^b^D^b^ mice, while Ly49A^+^, Ly49G2^+^ and NKG2A^+^ NK cells are educated in D^d^ mice [59]. The specific expression pattern endows each NK cell with a distinct specificity for a defined subset of MHC-I molecules [59]. Mice deficient in MHC-I expression will have NK cells that can become activated in response to cytokine stimulation or the recognition of strong activation signals, but are not educated by MHC-I to recognize the down-regulation of MHC-I on tumor cells [7,9]. By staining for these five inhibitory receptors and using Boolean gating, one can hone in on the function of specific NK cell subsets on different MHC-I backgrounds [9,59]. Interestingly, NK cells expressing the inhibitory receptors Ly49C, Ly49I and NKG2A proliferated significantly more in mice that expressed their cognate MHC-I alleles K^b^D^b^ compared to mice that expressed the D^d^ or no MHC-I molecules, as measured by BrdU-incorporation (Figure 4A,B). Conversely, NK cells that expressed the inhibitory Ly49A and Ly49G2 allele proliferated more in mice expressing their cognate D^d^ allele than in K^b^D^b^- or MHC-I-deficient mice (Figure 4B). NK cells not expressing any inhibitory receptor were proliferated equally in all three different MHC class I backgrounds (Figure 4B). Next, we addressed whether αGC treatment specifically boosts the function of NK cells expressing Ly49 receptors for their cognate MHC-I. In αNK1.1-antibody-activated NK cells from αGC-treated B6 mice, both CD107a and IFNγ expression was increased primarily in NK cells expressing only the educating Ly49C and Ly49I receptors (Figure 4C,D), while PMA/ionomycin-stimulated NK cells showed only minimal changes (Appendix A). Thus, we concluded that αGC treatment leads to NK cell proliferation of the educated NK cell pool and increases its functional capacity based on degranulation and cytokine expression.

### 2.5. Enhanced NK Cell Antitumor Response Following αGC-Treatment

Given the preferential proliferation and increased ex vivo functions of educated NK cells, we asked whether the αGC treatment also leads to an increase in NK-cell-mediated tumor killing in vivo. To be able to detect a change in the killing of both MHC-I-sufficient RMA and MHC-I-deficient RMA-S cells [1], we co-injected them together with spleen cells from αNK1.1 antibody-treated B6 mice as a reference population (Figure 5A). As predicted by the missing self-hypothesis, NK cells in untreated WT B6 and CD1d^−/−^ mice (educated NK cells) showed a stronger capacity to kill RMA-S cells than NK cells in MHC-I^−/−^ mice (uneducated NK cells). Killing of RMA-S (MHC-I^−^) cells was significantly increased in αGC-treated MHC-I^−/−^ mice and in B6 mice, but not CD1d^−/−^ mice treated with αGC (Figure 5B,C), highlighting the need for iNKT cell activation. αGC treatment was also sufficient to boost killing of RMA (MHC-I^+^) cells in both B6 and MHC-I^−/−^ mice (Figure 5B,D). Importantly, stimulation with PolyI:C, a dsRNA analogue commonly used for bystander activation of NK cells [53], did not increase killing of RMA and only minimally enhanced RMA-S killing in B6 mice or CD1d^−/−^ but induced elimination of both MHC-I^+^ and MHC-I^−^ tumor cells in MHC-I^−/−^ mice (Figure 5B–D). The enhanced response observed in MHC-I^−/−^ mice may reflect unspecific activation of uneducated NK cells. This is in line with previous studies showing that NK cells from MHC-I^−/−^ can react to MHC-I-deficient tumor cells even without education due to the recognition of ligands for the activation receptors on NK cells [9]. Taken together, our results suggest that αGC can increase the killing capacity towards cancer cells of both educated and uneducated NK cells.

### 2.6. αGC Boosts Missing Self-Response in MHC-I-Expressing Mice

Since treatment with αGC induces the proliferation and activation of educated NK cells and leads to increased tumor killing by both educated and uneducated NK cells, we asked whether this effect translates to an increase in “missing-self” rejection responses in vivo. To dissect true missing self-responses from the combinatorial input of activating and inhibitory ligands provided by tumor cells, we injected mice with spleen cells from MHC-I-sufficient and MHC-I-deficient mice. Strikingly, treatment with αGC enhanced missing self-reactivity towards MHC-I-deficient spleen cells for up to 7 days after treatment compared to PBS-treated mice (Figure 6A). This effect was dependent on the presence of iNKT cells, as there was no increase in missing self-reactivity in CD1d^−/−^ or Ja18^−/−^ mice, with both strains lacking a functional NKT cell compartment [60] (Figure 6B). Importantly, αGC-treatment did not enhance the missing self-response in MHC-I-deficient mice (Figure 6B). Hence, NK cell education is essential for the αGC-enhanced missing self-response towards non-malignant MHC-I-deficient cells.

We and others have previously shown that repeated injection of as little as 200 ng soluble αGC leads to blunted iNKT and T cell responses [32,45,46] in vivo. Here, we found that repeated αGC treatment led to an equally strong NK-cell-mediated missing self-response after two injections of αGC (Figure 6C). This suggests that the αGC-induced NK cell missing self-response is independent of the mechanism leading to iNKT-cell-dependent T cell-unresponsiveness following multiple stimulations with soluble αGC.

### 2.7. αGC-Loaded on Exosomes Induces Proliferation of Educated NK Cells and Increases Missing Self-Responses

Finally, we wondered whether we could combine boosting NK-cell-mediated missing self and antitumor immunity with a treatment that also augments adaptive antitumor immunity [47]. We have previously shown that DC-derived exosomes loaded with αGC and antigen (OVA) led to an increase in antigen-specific adaptive immune responses and antitumor activity [47]. Thus, we hypothesized that αGC-loaded exosomes could also enhance innate NK-cell-mediated responses. Here, soluble and exosomal αGC, used at corresponding doses, induced comparable NK cell activation and proliferation as measured by BrdU-incorporation and the up-regulation of activation markers CD69, KLRG1 (Figure 7A–C), and chemokine receptors CD62L, CXCR4, CX3CR1 and CCR5 (Appendix A) [47]. Importantly, exosomes loaded with protein antigen only (Exo(OVA)) did not induce NK cell activation (Figure 7A–C and Appendix A). Next, we tested whether Exo(αGC-OVA) or an equivalent dose of soluble αGC induced a similar bias in proliferation of educated NK cells. Exosomal αGC induced similarly strong NK cell proliferation as compared to soluble αGC (Figure 7C). Strikingly, we detected a dose-dependent proliferation bias of educated NK cells (Ly49C^+^Ly49I^+^NKG2A^+^) 7 days after exosome injection (Figure 7D). On the contrary, uneducated NK cells (Ly49A^+^Ly49G2^+^ or Ly49 receptor^−^NKG2A^−^) exhibited a dose-dependent decrease in proliferation-ratio, i.e., they proliferated less compared to the total pool of NK cells (Figure 7D). Finally, we assessed the impact of exosomal αGC on the missing self in vivo rejection capacity of NK cells. Similar to soluble αGC, Exo(αGC-OVA) but not Exo(OVA) increased the missing self-response in B6, but not in MHC class I- or CD1d-deficient mice (Figure 7E). Repeated injection of Exo(αGC-OVA) led to a similar increase in missing self-response without induction of anergy (Figure 7F). We conclude that the delivery of αGC on exosomes can boost the innate, NK-cell-dependent missing self-immunity in addition to its adjuvant effect on adaptive antitumor responses, reported previously [47].

## 3. Discussion

To avoid the various tumor escape mechanisms, which can be operational within the same tumor, it is essential to find treatments that can affect multiple arms of the immune response simultaneously. The interplay between different immune cells offers the possibility of cross-activation of different cell types. The current study shows that stimulation of iNKT cells with exosome-bound or soluble αGC, a natural CD1d ligand, leads to an increased functional capacity and preferential proliferation of educated NK cells in mice. Importantly, NK cells activated through iNKT cell-stimulation were better in killing tumor cells (both MHC-I^+^ and MHC-I^−^) and MHC-I-deficient non-tumor target cells, without breaking tolerance to self. Thus, by activating both adaptive [47] and innate immunity (this study), exosomes and the iNKT cell ligand αGC may thus offer potent tools for cancer immunotherapy.

Here, we demonstrate that αGC treatment leads to the proliferation of NK cells, with a preferential proliferation of educated NK cells, and furthermore increases their capacity for degranulation and cytokine production. In addition, our study shows that proliferation is, at least in part, dependent on CD8α^+^ cDC1. This is in line with cDC1s expressing the highest levels of CD1d as compared to other DC subsets such as cDC2s or plasmacytoid DCs (pDCs) [61,62]. In addition to their role in presenting lipid antigens on CD1d, DCs greatly enhance NK cell proliferation by providing stimulation through type I interferons, cytokines and cell-contact-dependent co-stimulation [58]. pDCs, as the main producers of type I IFN, and cDCs, as transpresenters of IL-15, play important roles in NK cell activation. iNKT cell stimulation with αGC leads to the release of numerous cytokines, including IL-2, IL-12 and IL-18, some of which are released by both cDCs and pDCs and all of which are known activators of NK cell proliferation and function [30,31,63,64,65,66]. For instance, IL-12 secreted by DCs enhances IFNγ production by NK cells, while IL-15 increases the proliferative response [56,67]. Moreover, DCs can present a membrane-bound form of IL-15 through which they can enhance NK cell functions in a contact-dependent manner [68]. DCs contribute to the differential activation of educated and uneducated NK cells, and we have previously shown that educated NK cells are endowed with a higher sensitivity to IL-15 [11]. We hypothesize that this property, at least in part, explains the proliferative advantage of educated NK cells in the current setting. Our data, including the lack of exhaustion of NK cell response after repeated αGC injection, support a dominant role for DCs in αGC-induced NK cell activation.

A number of studies have shown that in vitro cytokine stimulation can induce memory-like NK cells, with longevity and increased functional competence. In these settings, NK cells have been pre-activated with IL-12, IL-15 and IL-18, which led to increased IFNγ production and cytotoxic responses after the stimulation of NK cell receptors in vitro and after transfer to mice with established tumors in vivo [63,64,65]. Interestingly, while educated human NK cells showed the strongest increase in functional capacity, the KIR-negative subsets also elevated effector functions [63]. Our results, showing increased functional responses ex vivo and towards tumors in vivo, resemble those of cytokine-induced memory NK cells.

KLRG1^+^ NK cells play a role in the control of tumor growth [14], particularly in controlling metastasis formation [54]. The ligands of KLRG1, members of the cadherin family of adhesion molecules, are involved in intercellular adhesion that are expressed by many cell types and are up-regulated on inflammatory DCs and are involved in bidirectional signaling [69,70]. KLRG1 is primarily expressed by the most mature NK cells [14], but in our system, expression was already induced at the CD27^+^CD11b^−^ stage, which, to our knowledge, has not been described previously. The upregulation of KLRG1^+^ NK cells after αGC was dependent on NKT cells, as it was enhanced on NK cells from B6 or MHC-I^−/−^ mice, but not in CD1d^−/−^ mice. In contrast, when NK cells were activated with polyI:C, a TLR3 agonist that acts on NK cells via DCs, the up-regulation of KLRG1 was much lower, but could also be detected on NK cells from CD1d^−/−^ mice. However, in the absence of DCs after depletion in our CD11c.DOG mouse model, the up-regulation of KLRG1 after αGC treatment was abolished, demonstrating the importance of DCs also for αGC-induced KLRG1 upregulation.

While all NK cells were activated by αGC treatment, educated NK cells proliferated more and exhibited enhanced degranulation and cytokine production as compared to uneducated NK cell subsets. However, the mechanism for this increased proliferation and the functional response, primarily of the educated NK cells, is not clear. The current study describes an acute proliferative response in vivo where educated murine NK cells perform better than NK cells lacking self MHC-I receptors. In different studies of mice, e.g., after viral infection, the opposite is observed [71,72]. During CMV infection, the activation is mediated by a cognate receptor recognizing a viral product, which may be counteracted by the self-MHC-I-recognizing inhibitory receptors. However, educated NK cells may have an important role in controlling other viral infections [73,74]. In our system, the mechanism for the increased proliferation and functional response primarily of the educated NK cells is not clear, except that it is induced by iNKT cells, and is partially dependent on DCs. This may be similar to the development of adaptive NK cells in the human, which preferentially express inhibitory KIRs for self-MHC-I [75]. While NK cells lacking inhibitory receptors for self-MHC-I have been shown to proliferate more when stimulated with cytokines in vitro [76,77], this usually does not lead to an increase in Ly49-receptor-negative NK cells. Both differentiation and selective killing of NK cells lacking inhibitory MHC-I-specific receptors [76] seem to play a role in this. In our system, it is unclear whether the NK cells that express a self-specific inhibitory receptor (Ly49C, Ly49I, NKG2A) already expressed that receptor at the time of stimulation or whether they acquired it upon iNKT-cell-dependent activation and differentiation. Another possibility could be increased cell death by Ly49 receptor-negative NK cell subsets due to apoptosis or fractricide by educated NK cells, as was recently shown for human NK cells, where KIR^+^ NK cells kill KIR^−^ NK cells when competing for cytokines and growth factors in vitro [76]. Furthermore, a recent study has shown that KIR^−^NKG2A^−^ NK cells have a higher baseline transcription of pro-apoptotic factors than their educated counterparts [78]. However, we could not detect increased apoptosis of uneducated NK cells in our in vivo experiments (data not shown).

We have previously seen an anti-tumor effect of exosomes loaded with OVA in the B16-OVA model, an effect that was potentiated by αGC [47]. Here, we show that treatment of WT mice with soluble or exosomal αGC increased IFNγ secretion and degranulation upon receptor stimulation, specifically in educated NK cells ex vivo, and increased tumor rejection in vivo. However, we also observed an increase in tumor cell killing by NK cells from MHC-I-deficient mice after αGC treatment. NK cells in these mice showed an increased killing of tumor cells expressing MHC-I (RMA) and tumor cells lacking MHC-I (RMA-S). We speculate that this is due to a lowered activation threshold for all NK cells, as NK cells in MHC-I-deficient mice are less inhibited by interactions between Ly49 receptor and the cognate ligand [79]. It is a common misconception that NK cells lacking inhibitory receptors for self-MHC-I are unresponsive and cannot become activated. In fact, while NK cells from B2m^−/−^ do not reject B2m^−/−^ T cell blasts, they can respond to Yac-1 tumor cells [80]. In certain inflammatory conditions, such as CMV or influenza infections, uneducated cells even outperform their educated counterparts because they are less inhibited by the MHC-I of the infected cells [71,72]. While many cancers down-regulate MHC-I expression at some point during disease progression [51], many cancers do express MHC-I for a substantial part of their existence. In these situations, NK cells with no MHC-I-mediated inhibition are more effective at killing aberrant cells, as has been observed in mouse models lacking inhibitory receptors and in the NK-cell-mediated killing of neuroblastoma cells [79,81]. The importance of both NK cell subsets is furthermore highlighted by the fact that there seems to be a selective pressure to maintain a pool of both educated and uneducated cells within the NK cell population in mice [8] and humans [82].

Despite increased NK cell activation against both MHC-I^+^ and MHC-I^−^ tumors, it is critical that NK cells do not become autoreactive or that tolerance to self is not broken. This has been shown previously in studies using β2m^−/−^ or MHC-I-deficient mice, where the NK cells do not kill cells that lack MHC-I expression but react to MHC-I^−^ tumor cells [9,80]. According to the rheostat model for NK cell education, the responsiveness of NK cells is set based on the net signal result after balancing activating and inhibitory input [11]. In this model, the number and affinity of inhibitory MHC-I interactions with their cognate MHC-I ligands will determine the activation threshold. Thus, the more inhibitory input sensed by an individual NK cell, the stronger its responsiveness [5,83]. Therefore, NK cells sensing the lack of MHC-I in the absence of input from activating receptors will not react to healthy cells.

The increased expression of several chemokine receptors in our study could be an indication of the altered migration of NK cells. One possibility is that DCs produce the chemokines that attract NK cells, thus facilitating physical interaction. Once DCs and NK cells are in close proximity, DC-derived IL-12 or IL-15 presented by DCs could lead to the increased effector functions and proliferation [84]. The ability to interact with cognate MHC-I, together with higher levels of the adhesion receptor DNAM-1 on educated NK cells [85], could facilitate or prolong the interaction between DCs and NK cells, which would lead to a preferential presentation of IL-15 to specifically educated NK cells, explaining the increased proliferation of these subsets. Our experiments using CD11c.DOG mice show that proliferation is reduced in the absence of DCs. Taken together, the increase in chemokine receptors induced by αGC supports the positive effect of iNKT stimulation on NK cell anti-tumor effects.

It has previously been shown that repeated injections of soluble αGC lead to a long-term impairment of NKT cells’ ability to proliferate and produce IFNγ [86]. We have shown that iNKT-dependent T cell anergy could be overcome by linking αGC to exosomes [47]. Here, we show that repeated injections of αGC loaded exosomes, as well as αGC alone, does not induce unresponsiveness to missing self by NK cells, further supporting the use in cancer therapy. The possibility of combining αGC with protein antigens, different RNA species and ligands for innate immune receptors makes exosomes an attractive multi-potent delivery tool for cancer therapy. Combination therapy is increasingly discussed in the current cancer literature [87,88] and a report highlights the importance of activating iNKT cells with αGC when using DC-based immunotherapy against glioblastoma [89]. We propose exosomes as immunotherapy to target both MHC-I-expressing cancer cells via cytotoxic T cells and MHC-I-deficient cells via exosomal αGC NKT cell and bystander NK cell activation.

The current study shows that the stimulation of iNKT cells with exosome-bound or soluble αGC enhances the response of NK cells against cancer that have down-regulated MHC-I to escape T cell killing, and those that retain MHC-I expression in order to subvert recognition by NK cells. Thus, exosomes and the iNKT cell ligand αGC may offer potent tools for cancer immunotherapy.

## 4. Materials and Methods

### 4.1. Mice and Antibodies

C57Bl/6 mice (K^b^D^b^), beta-2-microglobulin-deficient mice (b2m^−/−^), MHC class I-deficient mice (K^b−/−^D^b−/−^, abbr. MHC-I^−/−^), MHC-I^−/−^ mice transgenic for the D^d^ allele (D^d^-single), CD1d^−/−^ or CD11c.DOG mice were bred and maintained under pathogen-free conditions at Karolinska Institutet’s animal facility. All animal experiments were approved by the Stockholm Regional Ethics committee (N418/12, N419/12, N70/15, N556/12 and N84/15).

### 4.2. Bone Marrow Derived Dendritic Cell (DC) Cultures and Exosome Production

Bone marrow cells were isolated from femur and tibia of female C57Bl/6 or CD1d^−/−^ mice and cultured in GM-CSF- and IL4-containing medium, as previously described [44]. After 6 days of culture, bone-marrow-derived DCs were pulsed with 300 µg/mL OVA (grade V, Sigma, St. Louis, MO, USA), and/or 100 ng/mL αGC (KRN-7000, Larodan Fine Chemicals, Malmö, Sweden) overnight (ON). Twenty-four hours later, cells were washed and replated in new medium containing exosome-depleted FCS [47], supplemented with 30 ng/mL LPS (Sigma, Stockholm, Sweden), and supernatants were harvested on day 9.We have shown that exosomes from DCs induce higher immune responses than MVs [90]; therefore, we isolated an exosome-enriched EV pellet by differential ultracentrifugation, as described before [44], called “exosomes” in this paper. First, supernatants were centrifuged 30 min at 3000× *g*, and then filtered using 0.22 µm cut-off filters (Nordic Biosite, Täby, Sweden). Remaining supernatants were then ultracentrifuged and exosomes washed in PBS at 100,000× *g* for 2 h 10 min. The final pellet was solubilised in PBS, aliquoted and frozen at −80 °C. Exosomal protein content was measured using a DC protein assay according to the manufacturer’s instructions (BioRad, Hercules, CA, USA).

### 4.3. In Vivo Proliferation

CD11c.DOG and WT mice were injected with DT to deplete DCs 24 h before αGC challenge. C57Bl/6 mice, MHC-I^−/−^D^d^-single mice or CD11c.DOG mice were injected on day 0 i.v. with 200 ng soluble αGC or 0.4, 4 or 40 µg of DC-exosomes. Mice were fed with 0.8 mg/mL 5-bromo-2′-deoxyuridine (BrdU, Sigma) supplemented with 2.5% sugar in drinking water either for 3 days or 7 days or in intervals from day 0 to 1, day 1 to 3, day 3 to 5 or day 5 to 7. Mice were sacrificed on days 1, 3, 5 or 7 and liver, lung and/or spleen were removed. Hepatic lymphocyte preparations and splenocyte single-cell suspensions were prepared as described previously [47]. BrdU-incorporation was measured using a BrdU staining kit (BDBiosciences, Eysins, Switzerland) according to the manufacturer’s instructions. Data were acquired using LSR II (BDBiosciences) and analyzed with FlowJo (Tree Star, Ashland, Oregon, USA). A proliferation index (index = %Brdu + cells in NK cells subset/%Brdu + cells of total NK cells) was calculated to normalize differences of BrdU uptake in the different mice, which may arise due to the different drinking habits of mice.

### 4.4. In Vivo NK Cell Activity

Spleens were mechanically disrupted into single-cell suspensions and erythrocytes were lysed. For tumor targets, RMA or RMA-S were grown in ascites of 400 rad irradiated mice, harvested after one week and washed extensively with PBS. Cells were then labeled with 0.5 µM (control cells: B6) or 0.8 µM (control cells: RMA) or 5 µM (target cells, MHC-I^−/−^ or RMA-S) of CFSE (Thermo Fisher, Waltham, Massachusetts, USA). Target and control cells were mixed in a 1:1 ratio and co-injected intravenously into mice, and the injection mix was analyzed for reference by flow cytometry [9]. For assessment of tumor cell killing, spleen cells of anti-NK1.1 antibody-treated B6 mice were labeled with BMQC violet (Invitrogen Molecular Probes) and co-injected with RMA and RMA-S cells to provide a reference population. At indicated time points, the spleens were harvested, erythrocytes depleted and the total or the relative percentages of cells in each population were measured by flow cytometry. The survival of target cells was estimated as relative survival, calculated as follows: (% target cells in sample/% control cells in sample)/(% target cells in inoculate/% control cells in inoculate) for spleen cell targets or (% target cells in sample/% reference cells in sample)/(% target cells in inoculate/% reference cells in inoculate) for tumor cell targets [9]. The results are expressed as survival ratio of remaining cells, target cells vs. control or target cells vs. reference. At least 3000 CFSE^+^ cells were acquired in each sample.

### 4.5. In Vitro Co-Culture Experiments

Spleens from C57Bl/6 and CD1d^−/−^ mice were homogenized into single-cell suspensions and red blood cells removed using lysis buffer according to the manufacturer’s protocol (Sigma). Cells were then stimulated for up to 48 h with different concentrations of αGC before expression of CD69 on NK1.1^+^, TCRb-, B220- NK cells were assessed using flow cytometry and a FACSVerse (BDBiosciences). In some experiments, splenocytes were first labeled with antibodies against CD11c, NK1.1, TCRb and B220 (all BDBiosciences) before DCs (defined as TCRb-, B220-, NK1.1-, CD11c+), NK cells (defined as TCRb-, B220-, NK1.1^+^, CD11c-) and iNKT cells (defined as TCRb+, B220-, NK1.1^+^, CD11c-) were sorted out using a FACSAria II (BDBiosciences). Sorted cells were then co-cultured in round-bottomed 96-well plates (Nunc) using 25,000 WT NK cells, 10,000 WT iNKT cells and 10.000 DCs from C57Bl/6 WT or CD1d^−/−^ mice per well, where indicated, in the presence or absence of 1 ug/mL αGC for up to 48 h. The expression of CD69 was assessed by flow cytometry as described above.

### 4.6. Degranulation and IFNγ Production Assay

In vitro degranulation and cytokine production assay was performed as previously described [9]. Briefly, spleen cells were stimulated with 20 µg/mL plate-bound anti-NKp46 (Mar-1; R&D Systems, Minneapolis, MN, USA), 20 µg/mL αNK1.1 antibody (MabTech, Nacka, Sweden) at indicated ratios for 4 h at 37 °C [9]. Cells were incubated in complete α-MEM medium supplemented with monensin (Golgi stop, BD Pharmingen), 10 µg/mL Brefeldin A (Sigma-Aldrich) and CD107a mAb (BDBiosciences). For positive controls, cells were stimulated with 20ng/mL PMA (Sigma) and 0.5 μg/mL ionomycin. Cells were harvested and stained for cell surface molecules and intracellularly for IFNγ (BDBiosciences) after fixation and permeabilization using the BD Pharmingen cytofix/cytoperm Plus Kit.

### 4.7. T-Distributed Stochastic Neighbor Embedding (tSNE) Analysis

Flow cytometry data were compensated and analyzed using FlowJo (Tree Star, v9.9.6). A total of 20,000 singlet lymphocytes were concatenated for normalization. A total of 20,000 singlet live NK cells (NKp46+CD3-) were concatenated for the analysis. The channel numbers were exported as comma-separated values. The data were then processed using R (version 3.5.2, 64 bit, The R Foundation, open source project). In brief, the data were normalized using the dscale function from DepecheR package (version 1.1.0, The R Foundation, open source) [55]. Barnes–Hut t-SNE was then performed using Rtsn function [91] on ten channels, namely NKp46, CD3, KLRG1, IFN-γ, CD107a, Ly49C, Ly49I, Ly49G2, Ly49A, and NKG2A. The plotting was done using DepecheR, gplots, and RColorBrewer packages. NK cells from B6 mice injected with αGC were defined manually based on tSNE regions that contain 10% or more cells from this group as compared to NK cells from untreated B6 mice (Figure 2F). These regions were then copied to other tSNE plots of individual groups in Figure 2G,H and Appendix A.

### 4.8. Statistical Analysis

Unpaired two-tailed student’s *t*-test, one-way or two-way ANOVA with Tukey’s correction for multiple testing was used for normally distributed data. Mann–Whitney or Kruskal–Wallis with Dunn’s correction was used for non-parametric data. Statistical analysis was performed using GraphPad software (GraphPad Inc., LaJolla, CA, USA).

## 5. Conclusions

In the current study, we show that stimulation of NKT cells with soluble as well as exosome-loaded αGC induces increased NK cell responses towards tumor cells. Educated NK cells show a superior proliferation response, and increased degranulation and cytokine production. However, both educated and uneducated NK cells are stimulated to kill cancer cells in vitro and in vivo.

Here, we propose exosomes as immunotherapy to target both MHC-I-expressing cancer cells via cytotoxic T cells and MHC-I-deficient cells via exosomal αGC NKT cell and bystander NK cell activation. The stimulation of this broad immunity suggests that exosomes could also be explored as treatment for chronic viral infections.

## Figures and Tables

**Figure 1 cancers-13-00298-f001:**
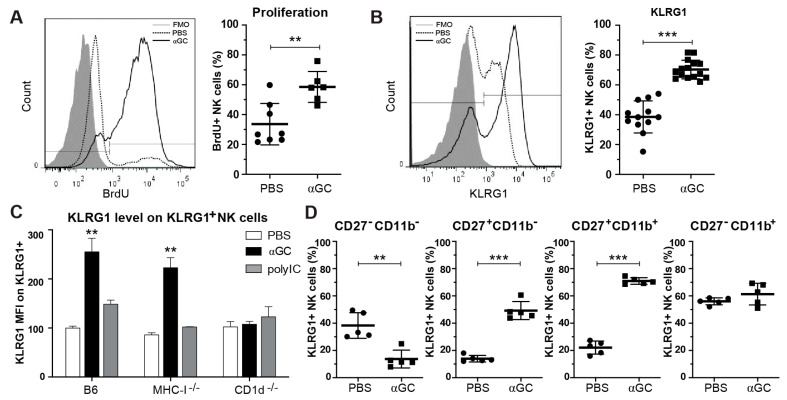
iNKT cell-dependent activation of NK cells by αGC in vivo. C57Bl/6 or CD1d^−/−^ mice were injected i.v. with PBS or soluble αGC. (**A**) B6 mice were fed BrdU in drinking water for 7 days and proliferation of NK cells was quantified by staining for BrdU (defined as BrdU^+^ of TCR-β^−^ or CD3^−^, NK1.1^+^ live lymphocytes). (**B**) Frequency of KLRG1^+^ NK cells and (**C**) mean fluorescence intensity (MFI) of KLRG1^+^ cells at day 3 after treatment of C57Bl/6, MHC-I^−/−^ or CD1d^−/−^ mice with PBS, αGC or polyI:C. MFI was normalized to expression level of PBS-treated B6 mice within each experiment. (**D**) Frequency of KLRG1^+^ NK cell subsets based on expression of maturation markers CD27 and CD11b. Data are pooled from 2–4 independent experiments and show results for (**A**) 6–8 mice per group (**B**) 12–16 mice per group (**C**) 4–9 mice per group for B6, 4–5 mice per group for MHC-I^−/−^ and CD1d^−/−^, (**D**) 5 mice per group. Statistical differences were calculated using unpaired two-tailed *t*-tests for (**A**,**B**,**D**) and Kruskal–Wallis test with Dunn’s correction for multiple testing within each group of mice compared to PBS-treated control (**C**) Error bars indicate mean ± SD and statistical significance is indicated as ** *p* < 0.01. *** *p* < 0.001.

**Figure 2 cancers-13-00298-f002:**
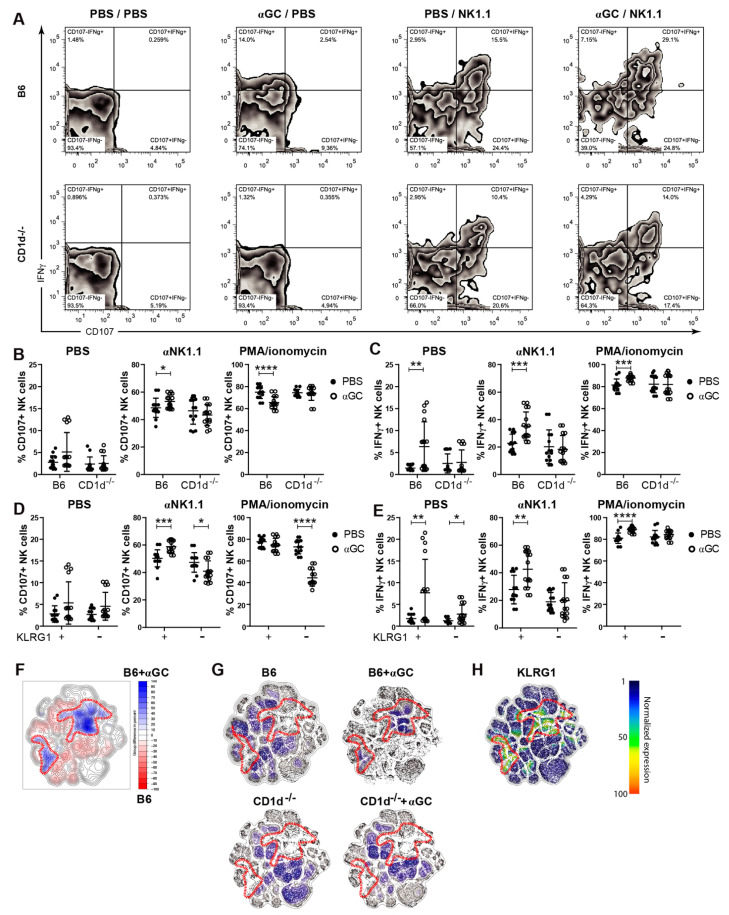
Activation of KLRG1+ natural killer (NK) cell subsets after in vivo stimulation with αGC**.** C57Bl/6 or CD1d^−/−^ mice were injected i.v., with PBS or soluble αGC. 3 days after in vivo stimulation, degranulation (by CD107 positivity) and cytokine production capacity (IFNγ production) were assessed. (**A**) Representative plots showing IFNγ (*y*-axis) vs. CD107 (*x*-axis) of unstimulated (PBS/PBS), in vivo αGC-stimulated and in vitro unstimulated (αGC/PBS), in vitro stimulated naïve (PBS/NK1.1) and in vivo plus in vitro stimulated (αGC/NK1.1) WT NK cells. Gated on single/live/CD3^−^NKp46^+^ cells. (**B**) CD107 and (**C**) IFNγ production of C57Bl/6 and CD1d^−/−^ NK cells stimulated with PBS (unstimulated), αNK1.1 or phorbol myristate acetate (PMA)/ionomycin. (**D**) CD107 and (**E**) IFNγ production of KLRG1^+^ vs. KLRG1^−^ NK cells of B6 mice, stimulated with PBS (unstimulated), anti-NK1.1 or PMA/ionomycin. (**F**–**H**) Flow cytometry data of mice of four groups (B6, B6 + αGC, CD1d^−/−^, CD1d^−/−^ + αGC) was analyzed with *t*-distributed stochastic neighbor embedding algorithm as previously described [55]. Combining B6 with B6 + αGC, areas with a 10% increase of cells were identified and manually gated (**F**) and subsequently applied to all remaining tSNE plots of different treatment groups individually (**G**). Data are pooled from 6 independent experiments with 14–16 mice per group (**A**–**E**) or from 3 independent experiments with 6 mice per group (**F–H**). Unpaired, two-tailed *t*-test (**B**–**E**) was used to determine statistical significance. Error bars indicate mean ± SD and statistical significance is indicated as * *p* < 0.05, ** *p* < 0.01. *** *p* < 0.001, **** *p* < 0.0001.

**Figure 3 cancers-13-00298-f003:**
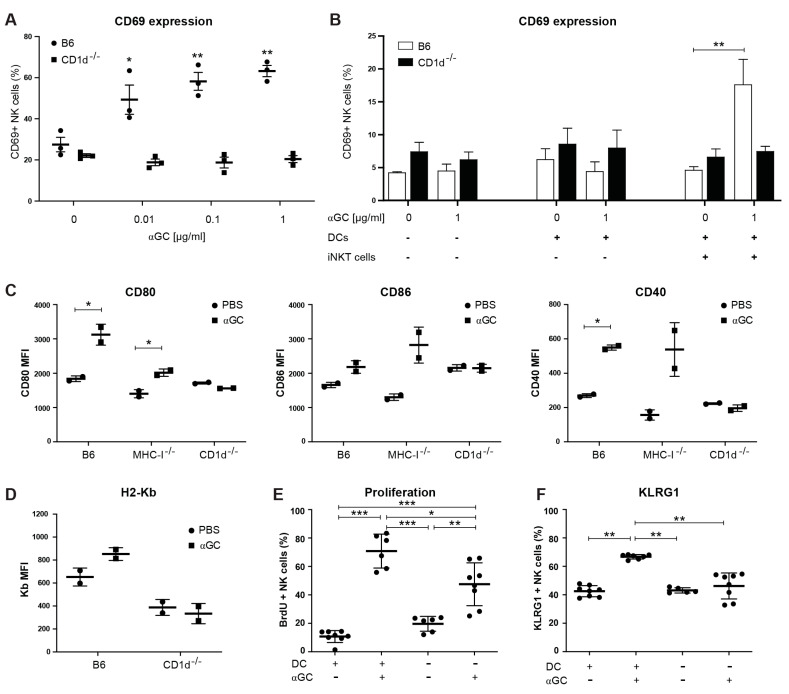
Dendritic cells (DCs) are activated by αGC in vivo and contribute to NK cell activation in vitro and in vivo. (**A**) Spleen cells of B6 and CD1d^−/−^ were isolated and cultured in vitro in the presence of absence of αGC (00.1, 0.1, 1 µg/mL), and CD69 expression measured on NK cells after 48h. (**B**) Spleen cells were isolated and sorted by magnetic bead sorting. NK cells were cultured alone, or in the presence of DCs and/or iNKT cells and in the presence or absence of αGC (1 µg/mL). CD69 expression was measured on NK cells after 24 h by flow cytometry. (**C**,**D**) C57Bl/6 mice were injected i.v. with PBS or with 200 ng soluble αGC. cDCs were stained at day 2 by excluding dead, CD3^+^NK1.1^+^CD19^+^Ter119^+^ cells, and then gated on CD11c^high^CD11b^low^CD8a^+^ cDCs. MFI of CD80, CD86 and CD40 (**C**) and H2-Kb (**D**). One representative experiment of 2 performed. (**E**,**F**) CD11c.DOG mice were treated with PBS (DC^+^) or DTR (DC^−^) for 7 days and received PBS or 200 ng of αGC and were fed BrdU via the drinking water from day 3. BrdU incorporation (**E**) or KLRG1 expression **(f)** was measured on day 7. Data are from 3 mice per group from one experiment (**A**,**B**), from 2 mice per group from one experiment (**C**,**D**), and 6–8 mice per group from two experiments (**E**,**F**). Statistical differences were calculated using one-way ANOVA with Tukey’s correction for multiple testing (**A**,**B**), where all groups where compared to the untreated B6 control group, unpaired two-tailed *t*-test (**C**,**D**), or one-way ANOVA with Tukey’s correction for multiple testing for (**E**,**F**). Error bars indicate mean ± SD and statistical significance is indicated as * *p* < 0.05, ** *p* < 0.01. *** *p* < 0.001.

**Figure 4 cancers-13-00298-f004:**
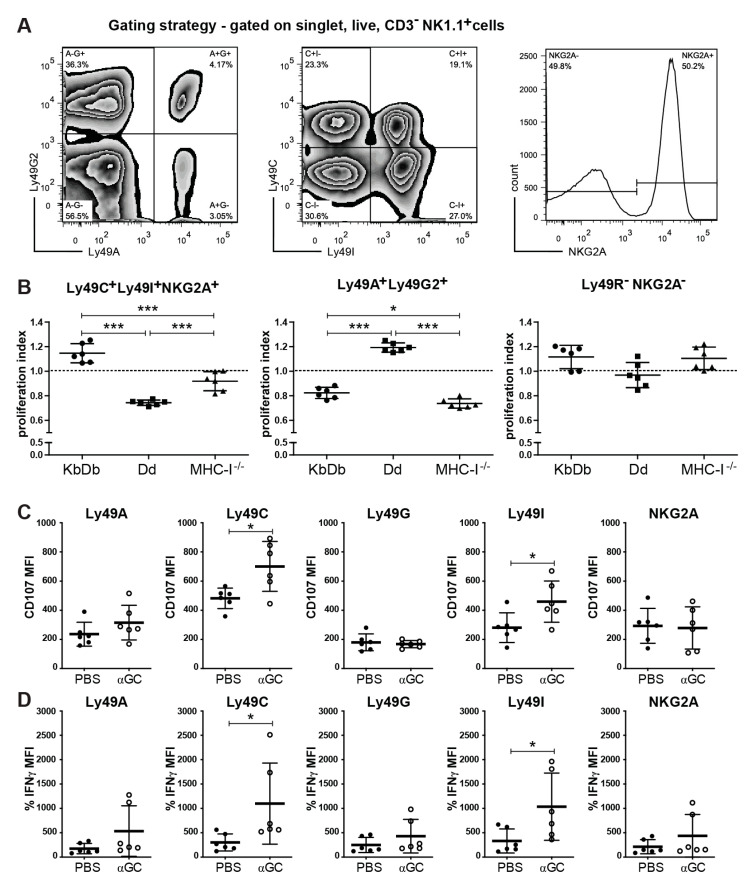
Selective proliferation of educated NK cell subsets after in vivo stimulation with αGC. C57Bl/6 mice (K^b^D^b^), MHC-I^−/−^ or MHC-I^−/−^ mice transgenic for the D^d^ allele (D^d^) were injected i.v., with PBS, or soluble αGC and fed with BrdU in drinking water for 7 days. (**A**) Gating strategy to show proliferation (BrdU incorporation) of NK (single, live, CD3^−^NK1.1^+^) cells with specific inhibitory receptors (Ly49G2 vs. Ly49A, Ly49C vs. Ly49I, NKG2A). (**B**) Proliferation ratio of NK cell subsets gated on the inhibitory receptors Ly49A, -C, -G_2_, -I and NKG2A using Boolean gating. The proliferation ratio was calculated as follows: (% BrdU^+^ NK cell subset/% BrdU^+^ total NK cells of the same mouse). (**C**,**D**) C57Bl/6 mice were injected i.v. with PBS or soluble αGC. 3 days after in vivo stimulation, degranulation (by CD107 positivity) and cytokine production capacity (IFNγ production) were assessed as for Figure 1. (**C**) CD107 and (**D**) IFNγ production of C56Bl/6 NK cell subsets based on the inhibitory receptors (Ly49G2 vs. Ly49A, Ly49C vs. Ly49I, NKG2A) stimulated with anti-NK1.1 antibody ex vivo. Data are pooled from 3 independent experiments with 6 mice per group (**B**–**D**). One-way ANOVA with Tukey’s post-test of correction for multiple testing (**B**) or unpaired, two-tailed *t*-test (**C**,**D**) was used.

**Figure 5 cancers-13-00298-f005:**
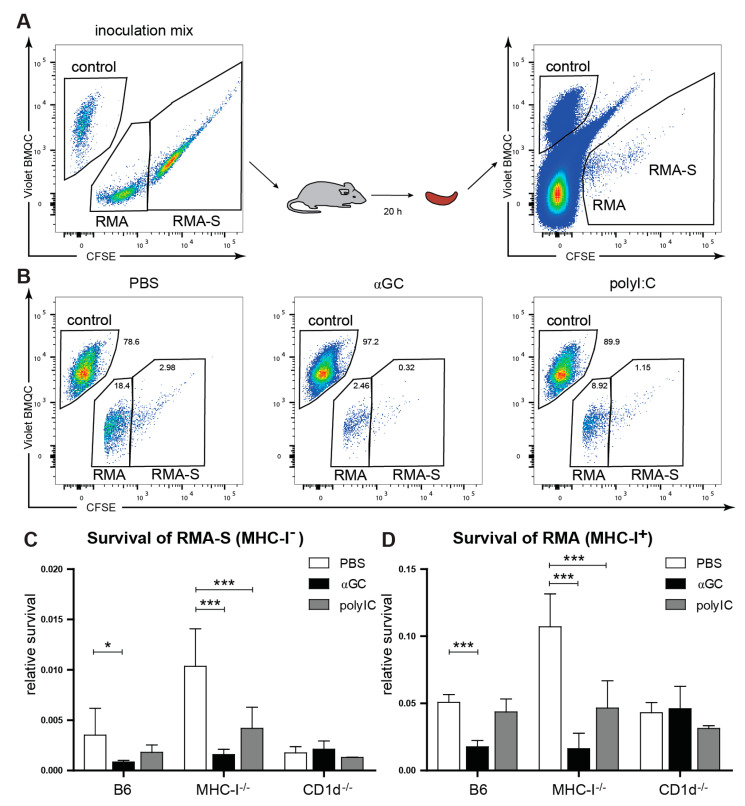
Increased anti-tumor responses after αGC-treatment. C57Bl/6, MHC-I^−/−^ or CD1d^−/−^ mice were injected i.v., with PBS, αGC or polyI:C. 3 days after in vivo stimulation, survival of MHC-I^+^ RMA (labeled with 0.5 uM CFSE) or MHC-I^−^ RMA-S (labeled with 5uM CFSE) tumor target cells was tested in reference to Violet BMQC-labeled MHC-I^+^ splenocytes (control) (**A**). Relative survival of MHC-I^+^ RMA or MHC-I^−^ RMA-S cells (**B**–**D**) was assessed by comparing the frequency of remaining target cells to an inert control cell population (NK1.1-depleted B6 spleen cells), as explained in Materials and Methods. (**B**) shows representative flow cytometry plots for B6 mice treated with PBS, αGC or poly-I:C. (**C**) The relative survival of remaining MHC-I^−^ target cells and (**D**) the relative survival of MHC-I^+^ target (**D**) are shown. (**C**,**D**) A total of 6–8 mice per group were tested (compiled from 2 independent experiments). Error bars denote SD. Significant differences were calculated within each mouse strain by one-way ANOVA with Dunnett’s correction for multiple testing and are denoted * *p* < 0.05, *** *p* < 0.001.

**Figure 6 cancers-13-00298-f006:**
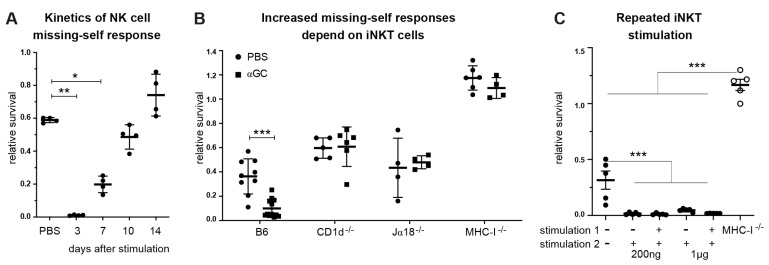
Increased missing self-responses after αGC-treatment. Mice were injected i.v. with PBS, or αGC. 3 to 14 days after in vivo stimulation, survival of target cells was tested by challenging the mice with a mix of CFSE-labeled B6 and β_2_m^−/−^ spleen cells. Remaining target cells were measured by flow cytometry. The survival ratio of remaining MHC-I^−^ (β_2_m^−/−^) target cells is shown, divided by remaining MHC-I^+^ (B6) control cells (relative survival). (**A**) Rejection capacity of NK cells in B6 mice was tested by injection of a mix of B6 and β_2_m^−/−^ spleen cells at d3, d7, d10 and d14 after in vivo treatment with αGC. (**B**) Missing self-responses of C57Bl/6, MHC-I^−/−^, CD1d^−/−^ or Jα18 were assessed at day 7 after in vivo αGC treatment. (**C**) B6 mice were treated with 200 ng or 1 µg of αGC (stimulation 1) and re-treated after 2 weeks (stimulation 2). Three days after the second treatment, the rejection capacity towards MHC-I-deficient spleen cells was assessed. Data are pooled from 2 independent experiments with 6 mice per group (**A**), 3 independent experiments with 9–12 mice per group for B6 and 4–6 mice per group for CD1d^−/−^, Jα18^−/−^ and MHC-I^−/−^ mice (**B**), and 2 independent experiments with 5 mice per group (**C**). One-way ANOVA comparing each group to PBS-treated control group with Dunnett’s correction for multiple comparisons (**A**) or Tukey‘s multiple comparison test (**C**) or unpaired, two-tailed *t*-test (**B**) was used to determine statistical significance. Error bars denote SD. Significant differences are denoted * *p* < 0.05, ** *p* < 0.01, *** *p* < 0.001.

**Figure 7 cancers-13-00298-f007:**
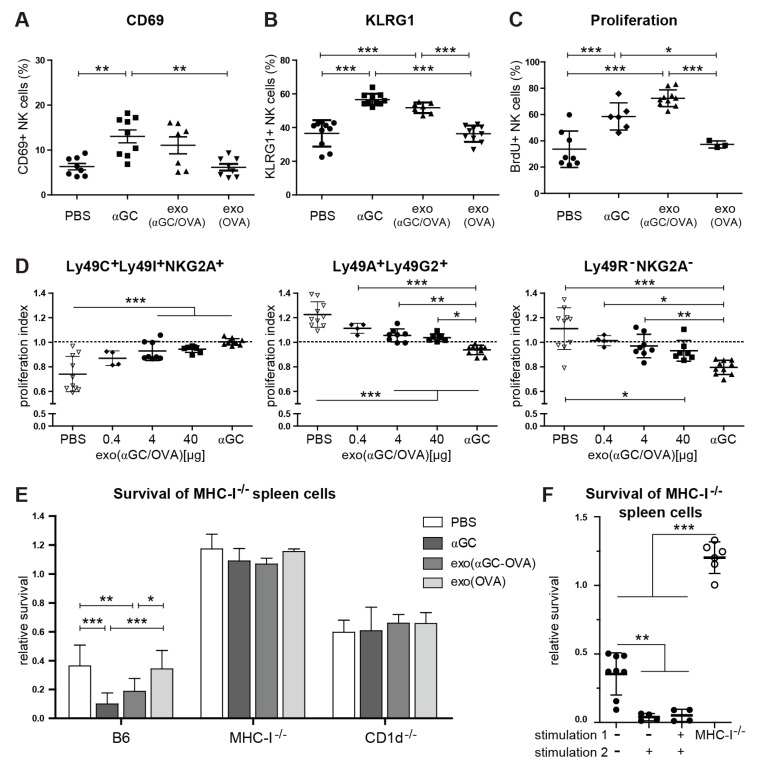
Exosomal αGC has comparable effects on NK cell activation, proliferation and missing self-function. C57Bl/6 mice were injected i.v., with PBS, 40 µg Exo(αGC-OVA) or soluble αGC (equivalent to the amount on 40 µg Exo(αGC-OVA) [47]. (**A**) Activation by CD69 and (**B**) KLRG1 expression was measured by flow cytometry on d7. (**C**,**D**) Mice were treated with PBS, 0.4, 4 or 40 µg exo(αGC-OVA) or with 200 ng soluble αGC and fed with BrdU in drinking water for 7 days. (**C**) Proliferation was assessed by measuring BrdU incorporation at day 7 on total NK cells. (**D**) NK cells were stained for inhibitory receptors (Ly49C/Ly49I/NKG2A/Ly49A/Ly49G_2_) and the proliferation index was assessed as described in Materials and Methods. The proliferation index of the Ly49C^+^Ly49I^+^NKG2A^+^Ly49A^−^Ly49G_2_^−^ (educated on self-MHC-I), the Ly49C^−^Ly49I^−^NKG2A^−^Ly49A^+^Ly49G_2_^+^ (not educated on self-MHC-I) and the Ly49C^−^Ly49I^−^NKG2A^−^Ly49A^−^Ly49G_2_^−^ (not educated on self-MHC-I) NK cell subsets are shown. (**E**,**F**) In vivo missing self-responses were measured by target cell elimination assay. (**E**) B6, MHC-I^−/−^ and CD1d^−/−^ mice were treated with 200 ng soluble αGC, 40 µg Exo(OVA) or 40 µg Exo(αGC-OVA) and challenged at day 7 with a mix of CFSE-labeled B6 (control) and β_2_m^−/−^ spleen cells (target). The ratio of surviving target cells was examined after two additional days. (**f**) B6 mice were treated twice with 40µg exo(αGC-OVA) at an interval of 2 weeks. 3 days after the second treatment (stimulation 2), the rejection capacity was assessed by target cell elimination of CFSE-labeled B6 (control) and β_2_m^−/−^ spleen cells (target). Data are pooled from 3 independent experiments (**A**–**D**) or from 2 independent experiments (**E**,**F**) with 7–10 mice per group (**A**,**B**), 3–10 mice per group (**C**), 4–10 mice per group (**D**), 4–12 mice per group (**E**) or 4–8 mice per group (**F**). Error bars denote SD. Significant differences were calculated with one-way ANOVA with Tukey’s correction for multiple testing and are denoted * *p* < 0.05, ** *p* < 0.01, *** *p* < 0.001.

## Data Availability

The data presented in this study are available on request from the corresponding author.

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
