# Peer review of "Soluble and Exosome-Bound α-Galactosylceramide Mediate Preferential Proliferation of Educated NK Cells with Increased Anti-Tumor Capacity"

_cancers, 2021, doi:10.3390/cancers13020298_

Round 1
Reviewer 1 Report
In this paper Wagner et al demonstrated that stimulation of NKT cells with soluble as well as exosome loaded αGC induces increased NK cell responses towards tumor cells. Although both educated and non-educated NK cells are stimulated to kill cancer cells in vitro and in vivo, MHC-I-educated NK cells show a superior proliferation response, and increased degranulation and cytokine production. This process requires DC to complete. The paper is of interest, the aims are clearly stated and Authors’ conclusions sustained by their results.
I have the following observation:
The Authors showed a series of important results, but no information is given on the mechanisms are taking place. As a matter of fact they should more precisely provide data on which type of dendritic cells are involved in the process, which cytokines are implicated and particularly how NK educated NK cells are preferentially activated. Since aCG mediated activation involves NKT and dendritic cells, it should of interest to the reader to recognize the mechanisms accounting for the specific NK cell activation
Author Response
We thank the reviewers for their comments that we feel helped us to improve the manuscript significantly. In addition, we are grateful to both the editor and reviewers for how swift our manuscript was reviewed and processed.
In this point-by-point response, we address the reviewer’s questions and suggestions. All references to specific lines or paragraphs in our responses refer to the revised document including track changes. We hope that the editor and the reviewers find our responses satisfactory.
Reviewer 1
In this paper Wagner et al demonstrated that stimulation of NKT cells with soluble as well as exosome loaded αGC induces increased NK cell responses towards tumor cells. Although both educated and non-educated NK cells are stimulated to kill cancer cells in vitro and in vivo, MHC-I-educated NK cells show a superior proliferation response, and increased degranulation and cytokine production. This process requires DC to complete. The paper is of interest, the aims are clearly stated and Authors’ conclusions sustained by their results.
I have the following observation:
The Authors showed a series of important results, but no information is given on the mechanisms are taking place. As a matter of fact, they should more precisely provide data on which type of dendritic cells are involved in the process, which cytokines are implicated and particularly how NK educated NK cells are preferentially activated. Since aCG mediated activation involves NKT and dendritic cells, it should of interest to the reader to recognize the mechanisms accounting for the specific NK cell activation
We thank the reviewer for asking this very relevant question. Indeed, it is important to report which type of DC is involved in the described processes. We have included our gating strategy for DCs (Supplemental Fig. 3) and have described the different DC subsets in the Introduction and the Results section in more detail. In addition, we have discussed the importance of CD8a+ cDC1 for both NKT cells and NK cells in the discussion. Concerning the cytokines, we provide a hypothesis as to the role of IL-15 in this setting. We have previously shown that educated NK cells show increased sensitivity towards IL-15, and as a result proliferate more.
These additions can be found in the Introduction (lines 98-101), section 2.3 of Results, and in the discussion (lines 797-813) and in Supplementary Figure 3b.
Reviewer 2 Report
Please review English.
The figures have a repetitive style. Please better represent your data so that a reader better remembers the information in your manuscript. Additionally, please do not use dynamite plots (bar and standard deviation; if you still wish to use this style in some instances please represent it in a style similar to Figure2B, although this also has it's flaws).
Please format the references in accordance with the MDPI style.
Overall I consider the article adds important information to the field. More than this, I could see that the authors have experience in this domain. As a suggestion, it would help a lot if you would have a person in your group that specialises in data visualisation as you could better present your results.
Author Response
We thank the reviewers for their comments that we feel helped us to improve the manuscript significantly. In addition, we are grateful to both the editor and reviewers for how swift our manuscript was reviewed and processed.
In this point-by-point response, we address the reviewer’s questions and suggestions. All references to specific lines or paragraphs in our responses refer to the revised document including track changes. We hope that the editor and the reviewers find our responses satisfactory.
Reviewer 2
Please review English.
The figures have a repetitive style. Please better represent your data so that a reader better remembers the information in your manuscript. Additionally, please do not use dynamite plots (bar and standard deviation; if you still wish to use this style in some instances please represent it in a style similar to Figure2B, although this also has it's flaws).
Please format the references in accordance with the MDPI style.
Overall I consider the article adds important information to the field. More than this, I could see that the authors have experience in this domain. As a suggestion, it would help a lot if you would have a person in your group that specialises in data visualisation as you could better present your results.
We thank the reviewer for his assessment of our manuscript. We have changed the style of the references to MDPI using endnote. In order to make the document more structured, these changes were accepted, and are not visible in the track changes.
In addition, we have changed most bar charts into scatter plots, unless this representation became too crowded. Furthermore, we have worked on the text and hope to have improved it substantially.
Reviewer 3 Report
some text polishing required
In this manuscript, Arnika Wagner et al. analyzed NK cell responses to alpha-galactosylceramide stimulation. Although this glycolipid is not physiologically relevant to mammals, it is well known as an activator of invariant NKT cells. Authors of this work demonstrate a profound influence of alpha-galactosylceramide on NK cells too. It induced NK cell proliferation and an increase in their functional activity with preferential effects on self-educated NK cells. Another interesting finding of the work is that in the composition of exosomes, alpha-galactosylceramide, along with the increase in missing self-response toward malignant cells, did not cause breaking of tolerance towards normal autologous cells. The data are novel and interesting. They are of a great interest in context of the development of new approaches for anti-cancer therapy.
The manuscript is well structured and the conclusions are convincingly supported by presented results. No big weaknesses were found in this work, but some text polishing is required.
I recommend this manuscript for publication in Cancers.
Author Response
We thank the reviewers for their comments that we feel helped us to improve the manuscript significantly. In addition, we are grateful to both the editor and reviewers for how swift our manuscript was reviewed and processed.
In this point-by-point response, we address the reviewer’s questions and suggestions. All references to specific lines or paragraphs in our responses refer to the revised document including track changes. We hope that the editor and the reviewers find our responses satisfactory.
Reviewer 3
In this manuscript, Arnika Wagner et al. analyzed NK cell responses to alpha-galactosylceramide stimulation. Although this glycolipid is not physiologically relevant to mammals, it is well known as an activator of invariant NKT cells. Authors of this work demonstrate a profound influence of alpha-galactosylceramide on NK cells too. It induced NK cell proliferation and an increase in their functional activity with preferential effects on self-educated NK cells. Another interesting finding of the work is that in the composition of exosomes, alpha-galactosylceramide, along with the increase in missing self-response toward malignant cells, did not cause breaking of tolerance towards normal autologous cells. The data are novel and interesting. They are of a great interest in context of the development of new approaches for anti-cancer therapy.
The manuscript is well structured and the conclusions are convincingly supported by presented results. No big weaknesses were found in this work, but some text polishing is required.
I recommend this manuscript for publication in Cancers.
We thank the reviewer for his kind assessment. We have worked extensively on the text and hope that the flow and the language is now improved.
Reviewer 4 Report
Wagner and colleagues showed the effects of αGC soluble or loaded in exosomes on NK cells. This is a potential interesting paper but but several issues should be taken into consideration carefully:
- only a small part of the work focuses on exosomes, so I suggets Authors to reformulate the abstract showing the novelties of the study even independently of the exosomes involvment
- I suggest Authors to improve the definition of iNK cells, educated NK cells and no educated NK cells
- I suggest Authors to improve the definition of specific inhibitory
receptors (Ly49G2 vs. Ly49A, Ly49C vs. Ly49I, NKG2A) in the context of NK cells. Even a less experienced reader must understand the data - I suggest Authors to improve the description of experiments in all Results section also following the above suggestions
- I suggest Authors to describe the three different MHC class I backgrounds and their meaning in the proposed experiments
- I suggest Authors to describe the meaning of PolyI:C administration in relation to experiments
- I suggest Authors to check the results described in "2.5. Enhanced innate antitumor immunity following αGC-treatment" paragraph
- Flow cytometry plots in Figure 5 are not clear, even if only representative of the whole experiment
- in the captions it is not clear what number of mice per experimental groups was used; moreover, I suggest to describe the meaning of challenge 1 and challenge 2 in figure 6 caption
- I suggest Authors to describe the source of exosomes extracted in results section
- I suggest Authors to clarify the meaning of "missing self-response"
- I suggest Authors to improve the discussion highlighting the novelty of the work
Author Response
We thank the reviewers for their comments that we feel helped us to improve the manuscript significantly. In addition, we are grateful to both the editor and reviewers for how swift our manuscript was reviewed and processed.
In this point-by-point response, we address the reviewer’s questions and suggestions. All references to specific lines or paragraphs in our responses refer to the revised document including track changes. We hope that the editor and the reviewers find our responses satisfactory.
Reviewer
Wagner and colleagues showed the effects of αGC soluble or loaded in exosomes on NK cells. This is a potential interesting paper but but several issues should be taken into consideration carefully:
We thank the reviewer for pointing out areas that we can improve. We have taken most suggestions and have described and defined the molecules and terms involved in more detail. Below, we have given a detailed response to each suggestion.
- only a small part of the work focuses on exosomes, so I suggets Authors to reformulate the abstract showing the novelties of the study even independently of the exosomes involvement
We have changed abstract and introduction, so that the results with soluble aGC are reflected better. The changes can be seen in the document including track changes in Abstract and Introduction.
- I suggest Authors to improve the definition of iNK cells, educated NK cells and no educated NK cells
We have defined our use of educated and uneducated NK cells in the introduction (introduction, lines 71-80). Furthermore, we have changed to a standardized use of educated vs uneducated NK cells throughout the paper.
- I suggest Authors to improve the definition of specific inhibitory
receptors (Ly49G2 vs. Ly49A, Ly49C vs. Ly49I, NKG2A) in the context of NK cells. Even a less experienced reader must understand the data
The inhibitory receptors are now better described, and we pointed out which inhibitory receptors are educating in the different mouse strains. This addition can be found in lines 575-585.
- I suggest Authors to improve the description of experiments in all Results section also following the above suggestions
We have not only introduced the inhibitory receptors and MHC-I alleles, but also described their interaction and the resulting education in the different mouse strains in the Results section. We thank the reviewer for making us aware that more info was needed for a general reader to understand.
- I suggest Authors to describe the three different MHC class I backgrounds and their meaning in the proposed experiments
We thank the reviewer for pointing this out. We had forgotten to introduce this complicated topic of Ly49 receptor – MHC allele interactions. We have now included a paragraph about specific inhibitory receptors and their cognate MHC-I alleles. This can be found in lines 575-585.
- I suggest Authors to describe the meaning of PolyI:C administration in relation to experiments
We have now added a short description of Poly I:C in the relevant sections in the Results (lines 229-230, 634-635).
- I suggest Authors to check the results described in "2.5. Enhanced innate antitumor immunity following αGC-treatment" paragraph
We have edited this section substantially and hope that it is now improved and that our methodology is comprehensible.
- Flow cytometry plots in Figure 5 are not clear, even if only representative of the whole experiment
In order to improve the comprehensibility of the figure and this assay in general, we have labeled the different cell populations in the figure as “control”, “RMA” and “RMA-S”.
- in the captions it is not clear what number of mice per experimental groups was used; moreover, I suggest to describe the meaning of challenge 1 and challenge 2 in figure 6 caption
We thank the reviewer for making us aware that we have used the word challenge for different meanings. We have changed the wording and are now using stimulation instead of challenge in sections 2.5 and 2.6, and in figures 5 and 6.
- I suggest Authors to describe the source of exosomes extracted in results section
We have added this in section 2.7.
- I suggest Authors to clarify the meaning of "missing self-response"
The term missing self-response is now defined in the introduction along with the relevant reference (lines 76-77).
- I suggest Authors to improve the discussion highlighting the novelty of the work
We have edited the discussion and added the novelty of our work.
Round 2
Reviewer 1 Report
The Authors responded adequately to reviewer's requests